# Oxygen vacancy-driven orbital multichannel Kondo effect in Dirac nodal line metals IrO$_2$ and RuO$_2$

Sheng-Shiuan Yeh[1,2,3], Ta-Kang Su[1], An-Shao Lien[1], Farzaneh Zamani[4], Johann Kroha [4], Chao-Ching Liao[1], Stefan Kirchner [5,6✉] & Juhn-Jong Lin [1,2,7✉]

Strong electron correlations have long been recognized as driving the emergence of novel phases of matter. A well recognized example is high-temperature superconductivity which cannot be understood in terms of the standard weak-coupling theory. The exotic properties that accompany the formation of the two-channel Kondo (2CK) effect, including the emergence of an unconventional metallic state in the low-energy limit, also originate from strong electron interactions. Despite its paradigmatic role for the formation of non-standard metal behavior, the stringent conditions required for its emergence have made the observation of the nonmagnetic, orbital 2CK effect in real quantum materials difficult, if not impossible. We report the observation of orbital one- and two-channel Kondo physics in the symmetry-enforced Dirac nodal line (DNL) metals IrO$_2$ and RuO$_2$ nanowires and show that the symmetries that enforce the existence of DNLs also promote the formation of nonmagnetic Kondo correlations. Rutile oxide nanostructures thus form a versatile quantum matter platform to engineer and explore intrinsic, interacting topological states of matter.

[1] NCTU-RIKEN Joint Research Laboratory, Institute of Physics, National Chiao Tung University, Hsinchu 30010, Taiwan. [2] Center for Emergent Functional Matter Science, National Chiao Tung University, Hsinchu 30010, Taiwan. [3] International College of Semiconductor Technology, National Chiao Tung University, Hsinchu 30010, Taiwan. [4] Physikalisches Institut and Bethe Center for Theoretical Physics, Universität Bonn, Nussallee 12, D-53115 Bonn, Germany. [5] Zhejiang Institute of Modern Physics and Department of Physics, Zhejiang University, Hangzhou 310027, China. [6] Zhejiang Province Key Laboratory of Quantum Technology and Device, Zhejiang University, Hangzhou 310027, China. [7] Department of Electrophysics, National Chiao Tung University, Hsinchu 30010, Taiwan. ✉email: stefan.kirchner@correlated-matter.com; jjlin@mail.nctu.edu.tw

Unconventional metallic states and the breakdown of the Landau Fermi liquid paradigm is a central topic in contemporary condensed matter science. A connection with high-temperature superconductivity is experimentally well established but the conditions under which these enigmatic metals form has remained perplexing[1]. One of the simplest routes to singular Fermi liquid behavior, at least conceptually, is through two-channel Kondo (2CK) physics[2–4]. Despite this long-standing interest, 2CK physics has thus far only been demonstrated to arise in carefully designed semiconductor nanodevices in narrow energy and temperature ($T$) ranges[5–8], while claims of its observation in real quantum materials are contentious (see "Discussion" section for details). More recently, the interest in Dirac and Weyl fermions within a condensed matter framework has led to the exploration of the effects of strong spin-orbit coupling (SOC) and of topological states which are rooted in a combination of time-reversal, particle-hole, and space-group symmetries[9,10]. While there has been considerable progress in understanding weakly correlated topological metals, only a few materials have been identified as realizing topological phases driven by strong electron correlations, which includes the Weyl–Kondo semimetals[11]. This raises the question if the 2CK counterpart of such a Weyl–Kondo semimetal, featuring an entangled ground state of the low-energy excitations of the 2CK effect with band-structure enforced Dirac or Weyl excitations, could at least in principle be stabilized. Exploring such a possibility, however, hinges on whether the 2CK effect can be stabilized at all in native quantum matter.

In this work we establish that oxygen vacancies ($V_O$'s) in the Dirac nodal line (DNL) materials $IrO_2$ and $RuO_2$ drive an orbital Kondo effect. $V_O$'s are prevalent in transition-metal oxides, including, e.g., $TiO_2$ and $SrTiO_3$, and their properties and ramifications have become central research topics as they can lead to an intricate entanglement of spin, orbital, and charge degrees of freedom[12–15]. The active degree of freedom in the orbital Kondo effect is not a local spin moment but a 'pseudospin' formed by orbital degrees of freedom[4]. In $IrO_2$ and $RuO_2$, the orbital Kondo effect is symmetry stabilized by the space-group symmetries of the rutile structure (Fig. 1). Both materials have been characterized as topological metals which feature symmetry-protected DNLs in their Brillouin zones[16,17]. This provides a link between the formation of the orbital Kondo effect and the presence of DNLs. In $IrO_2$ a nonmagnetic 2CK ground state ensues, while in $RuO_2$ the absence of time-reversal symmetry results in an orbital one-channel Kondo (1CK) effect.

The rutile structure type possesses mirror reflection, inversion, and a fourfold rotation ($C_4$) symmetry which enforce the presence of DNLs in the band structure of rutile oxides[10]. Some of these DNLs are protected from gapping out due to large SOC by the non-symmorphic symmetry of the rutile structure[18,19]. For $IrO_2$ and $RuO_2$ this has been recently confirmed by angle-resolved photoemission spectroscopy and band structure studies[16,17,19]. In the vicinity of $V_O$'s, this set of symmetries promotes the formation of the orbital 1CK and 2CK effect. The emergent Majorana zero mode that accompanies the formation of the 2CK effect is reflected in a singular excitation spectrum above the ground state which generates a $\sqrt{T}$-dependence of the resistivity $\rho(T)$ below a low-$T$ energy scale[20], the Kondo temperature $T_K$. This requires a well-balanced competition of two otherwise independent and degenerate screening channels and makes the 2CK effect extremely difficult to realize, especially in a natural quantum material[4,21,22]. If one channel dominates over the other, the low-$T$ behavior will be that of conventional fermions. If the 2CK state arises out of orbital Kondo scattering, magnetic-field ($B$) independence is expected

for field strengths well above $T_K$ as long as $g\mu_B B \ll W$, where $g$ is the Landé factor, $\mu_B$ is the Bohr magneton, and $W$ is the conduction electron half-bandwidth. Our study is based on rutile ($MO_2$, $M =$ Ir, Ru) nanowires (NWs) which allow us to combine a high degree of sample characterization with an exceptional measurement sensitivity while probing material properties in the regime where the characteristic sample dimension is much larger than the elastic electron mean free path (cf. Supplementary Note 3). That is, we are concerned with weakly disordered, diffusive metals which are three-dimensional (3D) with respect to the Boltzmann transport, whereas strong correlation effect causes a resistivity anomaly at low $T$. Table 1 lists the relevant parameters for the NWs studied in this work.

## Results

**Oxygen vacancies in transition-metal rutiles $MO_2$.** In Fig. 1a, the vicinity of an $V_O$, denoted $V_{O1}$, is shown. The metal ions surrounding $V_{O1}$, labeled $M1$, $M2$, and $M3$, form an isosceles triangle (Fig. 1b). For the sites $M1$ and $M2$, an almost perfect $C_{4v}$ symmetry exists which implies a corresponding degeneracy associated with the two-dimensional irreducible representation of $C_{4v}$, see Fig. 1c and Supplementary Note 4. In the pristine system, the metal ions are surrounded by oxygen octahedra anchored around the center and the corners of the tetragonal unit cell. The $\pi/2$ angle between adjacent octahedra leads to a fourfold screw axis symmetry. This non-symmorphic symmetry not only protects DNLs in $IrO_2$ against SOC-induced splitting[17,19]. It has also been linked to the high electrical conductivity of $IrO_2$ (ref. [10]) and, as we find, is in line with the strong tendency to localize electrons near $V_O$'s required for the formation of orbital Kondo correlations. Moreover, the fourfold screw axis symmetry ensures that the $C_4$ rotation axes centered at the sites $M1$ and $M2$ near $V_{O1}$ are not parallel ($\hat{z}' \nparallel \hat{z}$, see Fig. 1d). This enhances the phase space for the orbital Kondo effect over orbital order linking sites $M1$ and $M2$ (see also Supplementary Note 5).

**Experimental signatures of orbital 2CK effect in $IrO_2$ NWs.** Now we turn to our experimental results which, to the best of our knowledge, demonstrate the most convincing realization of the long searched orbital 2CK effect in a solid. Fig. 2 demonstrates the formation of an orbital 2CK effect in $IrO_2$ NWs. We find that as $T$ decreases from room temperature to approximately a few Kelvin, $\rho(T)$ decreases in all $IrO_2$ NWs, as expected for typical metallic behavior (cf. Supplementary Note 2). However, below $T \sim 20$ K, $\rho(T)$ displays a $\sqrt{T}$ increase of the $\rho(T)$ upon lowering $T$ over almost two decades in $T$(!), until a deviation sets in at ~0.5 K. We performed systematic thermal annealing studies to adjust the oxygen contents in the NWs, which indicate that the anomalous low-$T$ transport properties are driven by the presence of $V_O$'s (ref. [23] and Supplementary Note 1). This is exemplified in Fig. 2. The top left inset shows a scanning electron microscopy image of NW A. In the oxygenated NW 3 which should contain a negligible amount of $V_O$'s, $\rho(T)$ decreases monotonically with decreasing $T$, revealing a residual resistivity, $\rho_{B0}$, below ~4 K (top right inset). In contrast, in NWs A, B1 and B2 which contain large amounts of $V_O$'s, $\rho(T)$ increases with decreasing $T$, manifesting a robust $\rho \propto \sqrt{T}$ law between ~0.5 and ~20 K. The slope of NW B2 is smaller than that of NW B1, which indicates a decrease in the number density of oxygen vacancies ($n_{V_O}$) due to prolonged aging (for about 5 months) in the atmosphere. The data explicitly demonstrate that the $\rho \propto \sqrt{T}$ behavior is independent of $B$ up to at least 9 T. The observed behavior is consistent with the 2CK effect as indicated by the straight solid lines

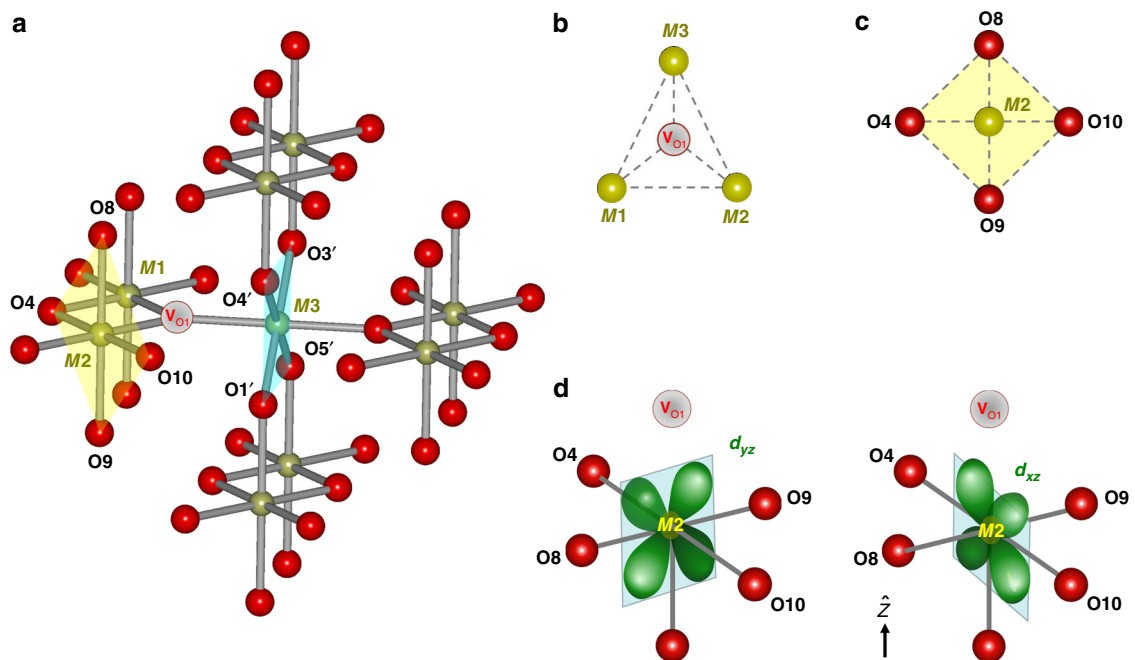

**Fig. 1 Atomic arrangement around an oxygen vacancy in $MO_2$ rutile structure. a** Schematics for $MO_2$ in the rutile structure. The olive and red spheres represent transition-metal ions $M^{4+}$ and oxygen ions $O^{2-}$, respectively. $V_{O1}$ represents an oxygen vacancy. **b** The metal ions $M1$, $M2$, and $M3$ surrounding $V_{O1}$ form an isosceles triangle. **c** The four oxygen ions surrounding $M2$, labeled O4, O9, O10, and O8, form an almost perfect planar square (while O1', O5', O3', and O4' only form a rectangle, cf. Supplementary Note 4 for details). **d** The $d_{xz}$ and $d_{yz}$ orbitals at $M2$ next to $V_{O1}$, with $\hat{z}$ perpendicular to the O4, O8, O10, and O9 plane, remain essentially degenerate as a result of mirror and $C_4$ rotation symmetry around $M2$. (Due to the non-symmorphic rutile structure, $\hat{z} \nparallel \hat{z}'$, where $\hat{z}'$ is parallel to the $C_4$ axis at $M1$.).

**Table 1 Relevant parameters for $MO_2$ NWs.**

| NW | $d$ | $\rho$(300 K) | $\rho_{B0}$ | $\ell$(10 K) | $D$(10 K) | $\rho_{K0}$ | $T_K$ | $n_{V_O}$ | $n_{V_O}/n_O$(%) |
|---|---|---|---|---|---|---|---|---|---|
| $IrO_2$ A | 130 | 147 | 109 | 2.5 | 4.2 | (0.65) | (20) | ~1.9 × 10²⁵ | ~0.031 |
| $IrO_2$ B1 | 190 | 104 | 73.9 | 3.7 | 6.2 | (0.72) | (20) | ~2.2 × 10²⁵ | ~0.036 |
| $IrO_2$ B2 | 190 | 106 | 75.0 | 3.6 | 6.0 | (0.45) | (20) | ~1.4 × 10²⁵ | ~0.023 |
| $RuO_2$ A | 53 | 193 | 122 | 2.2 | 4.0 | 0.94 | 3.0 | ~1.5 × 10²⁵ | ~0.025 |
| $RuO_2$ B | 67 | 163 | 120 | 2.3 | 4.2 | 14 | 12 | ~2.3 × 10²⁶ | ~0.38 |
| $RuO_2$ C | 54 | 589 | 434 | 0.63 | 1.2 | 17 | 69 | ~2.7 × 10²⁶ | ~0.44 |
| $RuO_2$ D | 120 | 245 | 160 | 1.7 | 3.1 | 7.0 | 80 | ~1.1 × 10²⁶ | ~0.18 |
| $RuO_2$ E | 47 | 761 | 587 | 0.47 | 0.9 | 30 | 7.0 | ~4.8 × 10²⁶ | ~0.79 |

Diameter $d$ is in nm, room-temperature resistivity $\rho$(300 K), residual resistivity $\rho_{B0}$, and Kondo resistivity in the unitary limit $\rho_{K0}$ are in $\mu\Omega$ cm, the electron mean free path $\ell$(10 K) is in nm, the electron diffusion constant $D$(10 K) is in cm² s⁻¹, the Kondo temperature $T_K$ is in K, and the number density of oxygen vacancies $n_{V_O}$ is in m⁻³. $n_O$ denotes the oxygen atom number density in the rutile structure. In all 4-probe configuration for transport measurements, the length between the two voltage probes is ~1 μm. The $\ell$(10 K) and $D = 1/[\rho e^2 N(E_F)] = \frac{1}{3} v_F \ell$ values are calculated through the free-electron model, where $N(E_F)$ is the density of states at the Fermi energy, and the Fermi velocity $v_F \approx 5.0 \times 10^5$ and $5.5 \times 10^5$ m s⁻¹ in $IrO_2$ and $RuO_2$, respectively. For each $IrO_2$ NW, we have empirically taken the $\rho_{K0}$ value to be the maximum value of the measured Kondo resistivity at ~0.5 K and $T_K \simeq 20$ K. These values are listed in parentheses. $IrO_2$ NW B has been measured twice before and after oxygenation in air and labeled B1 (first measurement) and B2 (second measurement).

which are linear fits to the 2CK effect calculated within the dynamical large-$N$ method (cf. Supplementary Note 5), with $n_{V_O}$ as an adjustable parameter (see Table 1 for the extracted values and Supplementary Notes 5 and 6 for the extraction method).

**Ruling out the 3D electron–electron interaction (EEI) effect.** To complicate matters, the EEI effect in 3D weakly disordered metals generically leads to a $\sqrt{T}$ term in $\rho(T)$ at low $T$ (refs. [24,25]). Unambiguously establishing that $\rho(T) \sim \sqrt{T}$ indeed originates from 2CK physics thus requires a proper analysis of the EEI effect of the charge carriers. For example, for the NW B1 with $\rho_{B0} = 74$ μΩ cm and the electron diffusion constant $D \simeq 6.2$ cm² s⁻¹, the 3D EEI effect would predict a largest possible resistance increase of $\Delta\rho/\rho \simeq 2.8 \times 10^{-4}$ as $T$ is cooled from 20 to 1 K. Experimentally, we have observed a much larger resistance increase of $5.1 \times 10^{-3}$. Furthermore, the 3D EEI effect would predict similar values for the

magnitude of the low-$T$ resistivity increase in NWs B1 and B2 to within ≈3%, due to their $\rho_{B0}$ values differing by ≈1% (Table 1). This is definitely incompatible with our observation of a ≈50% difference. In addition, we find a deviation from the $\sqrt{T}$ behavior at ~0.5 K. If the $\sqrt{T}$ anomaly were caused by the EEI effect, no such deviation should occur (see Supplementary Note 3 for an in-depth analysis of the EEI effect and its 3D dimensionality in our $MO_2$ NWs).

**$V_O$-driven orbital Kondo scattering in $MO_2$.** For $IrO_2$ the valency of the transition-metal ion $M$ is close to the nominal valence of +IV in $MO_2$ (ref. [26]). Each $V_O$ generates two defect electrons due to charge neutrality. To minimize Coulomb interaction, the defect electrons will tend to localize at different $M$ ions in the vicinity of the $V_O$. In $IrO_2$ this results in a nonmagnetic $5d^6$ ground state configuration of the Ir ions. For the electron

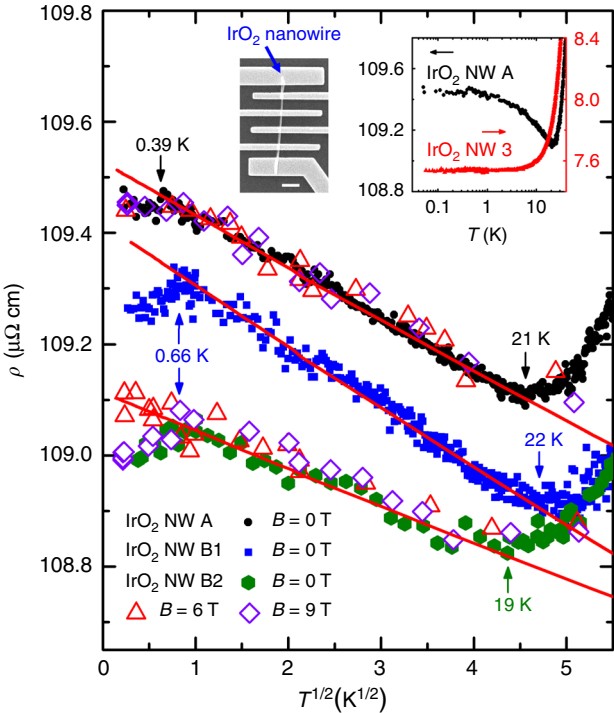

**Fig. 2 Orbital 2CK resistivity of IrO₂ NWs.** $\rho$ versus $\sqrt{T}$ for IrO₂ NWs A, B1 and B2 in magnetic fields $B = 0$, 6, and 9 T, as indicated. For clarity, the data of NWs B1 and B2 are shifted by 34.7 and 33.6 $\mu\Omega$ cm, respectively. A $\rho \propto \sqrt{T}$ law, which is $B$ independent, is observed between ~0.5 and ~20 K in all three NWs. The straight solid lines are linear fits to the 2CK resistivities calculated by the dynamical large-$N$ method (see text). Top left inset: a scanning electron microscopy image of NW A. The scale bar is 1 $\mu$m. Top right inset: Low-$T$ $\rho(T)$ curves of NW A and a reference, oxygenated NW 3 (diameter $d = 330$ nm, $\rho(300 \text{ K}) = 124$ $\mu\Omega$ cm).

localizing on ion $M2$ or $M1$ (Fig. 1a), the symmetry of the effective potential implies the almost perfect degeneracy of the orbitals $d_{xz}$ and $d_{yz}$ as defined in Fig. 1d. It is this orbital degeneracy that drives the orbital 2CK effect in IrO₂ where the $d_{xz}$ and $d_{yz}$ form a local pseudospin basis, while the spin-degenerate conduction electrons act as two independent screening channels. Group theoretical arguments ensure that the exchange scattering processes between conduction electrons and pseudospin degree of freedom have a form compatible with the Kondo interaction[27] (cf. Supplementary Note 5). Deviations from perfect symmetry which act as a pseudo-magnetic field are expected to become visible at lowest $T$. This explains the deviations from the $\sqrt{T}$ behavior observed below ~0.5 K in Fig. 2. If the two defect electrons localize at sites $M1$ and $M2$, a two-impurity problem might be expected which could lead to inter-site orbital order between the two defect electrons[28]. The non-symmorphic rutile structure, however, ensures that the $C_4$ rotation axes centered at the sites $M1$ and $M2$ are not parallel. This together with the local nature of the decomposition provided in Supplementary Eq. (3) (see Supplementary Note 5) favor local orbital Kondo screening in line with our observation. These conclusions are further corroborated by demonstrating tunability of the orbital 2CK effect to its 1CK counterpart.

**Experimental signatures of orbital 1CK effect in RuO₂ NWs.** RuO₂ is also a DNL metal with the same non-symmorphic symmetry group as IrO₂ but weaker SOC. In contrast to IrO₂, it lacks time-reversal symmetry[19,29]. Based on the analysis for IrO₂, we expect that $V_O$'s in RuO₂ will drive an orbital 1CK effect. This

is indeed borne out by our transport data on RuO₂ NWs. Fig. 3a shows the $T$ dependence of the time-averaged Kondo resistivity $\langle \rho_K \rangle$ for NW C, where $\rho_K(T) = \rho(T) - \rho_{B0}$, and $\langle \dots \rangle$ denotes averaging. (RuO₂ NWs often demonstrate temporal $\rho$ fluctuations. Details can be found in Supplementary Note 2.) At low $T$, $\langle \rho_K \rangle$ follows the 1CK form[30]. The inset demonstrates the recovery of a Fermi-liquid ground state with its characteristic $\langle \rho_K \rangle \propto T^2$ behavior below ~12 K and unambiguously rules out the 3D EEI effect. Fig. 3b shows $\rho(T)$ of NW E in $B = 0$ and 4 T. For clarity, the $B = 0$ data (black symbols) are averaged over time, while the $B = 4$ T data (red symbols) are non-averaged to demonstrate the temporal fluctuations of the low-$T$ $\rho(T)$ (ref. [31]). Note that, apart from the aforementioned much smaller resistance increase as would be predicted by the 3D EEI effect compared with the experimental results in Fig. 3a, b, no $\sqrt{T}$ dependence is detected here. In fact, the low-$T$ resistivity anomalies conform very well to the 1CK scaling form for three decades in $T/T_K$ (Fig. 4a). Thus, the 3D EEI effect can be safely ruled out as the root of the observed low-$T$ resistivity anomalies in RuO₂ NWs.

As a further demonstration of the $B$-field independence, we present in Fig. 3c $\rho(T)$ data for NW A in magnetic fields of strength $B = 0$, 3, and 5 T. With $T_K^A = 3$ K, NW A has the lowest $T_K$ among NWs A–E (Table 1). The data between 50 mK and 10 K, corresponding to $T/T_K = 0.017$–3.3, can be well described by the 1CK function (solid curve). The dash-dotted curves depict the magnetoresistance predicted by the spin-$\frac{1}{2}$ Kondo impurity model[30] with $g\mu_B B/k_B T_K = 1.0$, 2.0, and 4.1, as indicated, where $k_B$ is the Boltzmann constant. Our experimental data clearly demonstrate $B$ independence, ruling out a magnetic origin of this phenomenon.

We remark on the relation between the residual resistivity $\rho_{B0}$ and the concentration of orbital Kondo scatterers $n_{V_O}$ extracted from $\rho_{K0}$, the Kondo contribution to the $\rho(T \to 0)$ (see Supplementary Note 6), for RuO₂ NWs. With the exception of NW B, our data indicate an approximately linear relation between $n_{V_O}$ and $\rho_{B0}$ (Table 1 and Supplementary Fig. 3). It is not unexpected that the approximately linear relation between $\rho_{B0}$ and $n_{V_O}$ holds for larger impurity concentrations, corresponding to larger values of $\rho_{B0}$ as all defects, screened dynamic and static defects, contribute to $\rho_{B0}$. This relation strongly demonstrates that the low-$T$ resistivity anomalies are indeed due to $V_O$-driven orbital Kondo effect. (We focus on RuO₂ NWs because of the larger number of samples with a larger variation of $\rho_{B0}$ values compared with IrO₂ NWs).

**Comparison of 2CK and 1CK $\rho(T)$ curves.** Figure 4a demonstrates that $\langle \rho_K \rangle / \rho_{K0}$ for RuO₂ NWs follow the universal 1CK scaling over three decades in $T/T_K$ while $T_K$ ranges from 3 to 80 K! To further substantiate the subtle but distinct differences between the $\sqrt{T}$ dependence of the 2CK behavior in IrO₂ NWs from the 1CK scaling form, we plot $\langle \rho_K \rangle / \rho_{K0}$ as a function of $\sqrt{T/T_K}$ for IrO₂ NWs A and B1, together with RuO₂ NWs B–E and the 1CK function, in Fig. 4b, c, respectively. (The value for $\rho_{K0}$ was identified with the maximum values of the measured $\rho_K(T)$ anomalies.) Fig. 4d illustrates that a dilute system of 2CK scattering centers immersed in a metallic host indeed displays a $\sqrt{T}$ term in its low-$T$ $\rho(T)$. This $\sqrt{T}$ power-law behavior is determined by the leading irrelevant operator near the 2CK fixed point[32] and captured by the dynamical large-$N$ method[33–35].

**Discussion**
Despite the ubiquitous appearance of magnetic Kondo scattering in real quantum materials[36], no convincing demonstration of the orbital Kondo effect[37] or the 2CK effect[22,38] exists. Many claims rest on a model of two-level systems immersed in a metallic host as a

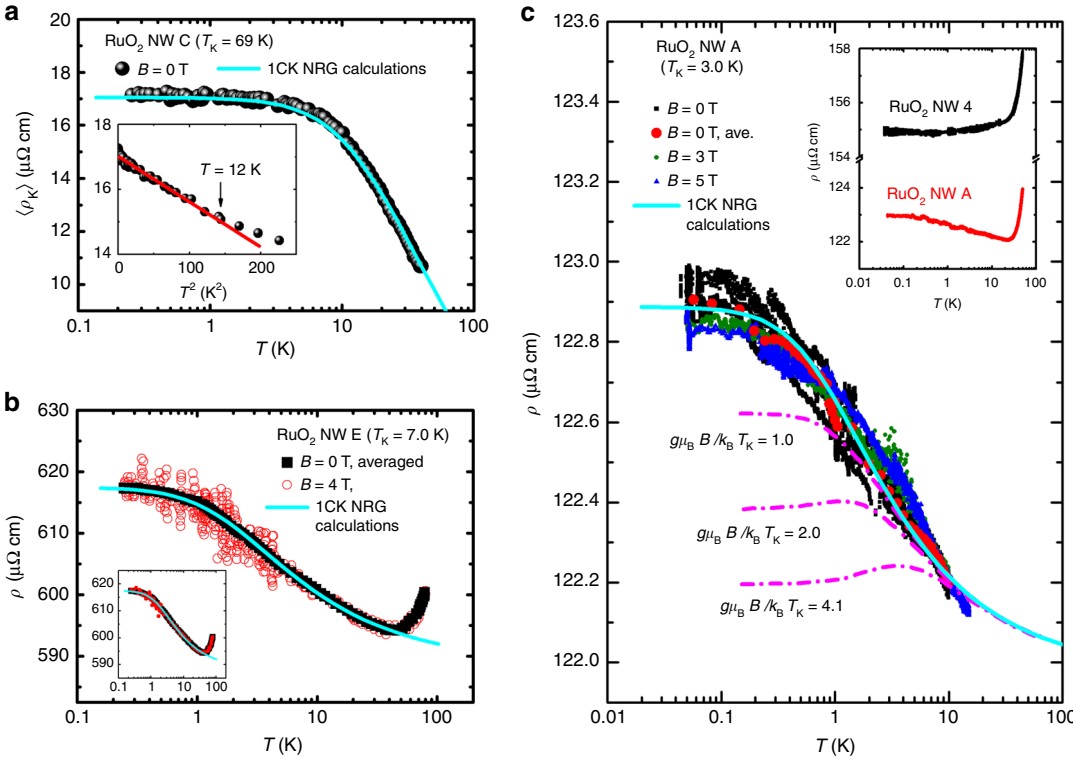

**Fig. 3 Orbital 1CK resistivity of RuO₂ NWs. a** Time-averaged Kondo resistivity $\langle\rho_K\rangle$ versus log $T$ for NW C. The straight line in the inset, which shows a low-$T$ zoom-in, is a guide to the eye. **b** $\rho$ versus log $T$ in $B = 0$ and 4 T for NW E. For clarity, the $B = 0$ data are time-averaged, while the 4-T data are non-averaged to demonstrate the temporal resistivity fluctuations at low $T$. The inset shows the time-averaged $B = 4$ T data (red symbols), which closely overlap the $B = 0$ data. **c** $\rho$ versus log $T$ for NW A in $B = 0$, 3, and 5 T. Occasional resistivity jumps, or random telegraph noise, are observed. The dash-dotted curves depict the magnetoresistance predicted by the spin-$\frac{1}{2}$ Kondo impurity model (see text). Note that the experimental data are independent of $B$. Inset: Low-$T$ $\rho(T)$ curves of NW A and a reference, oxygenated NW 4 ($d = 150$ nm, $\rho(300\text{ K}) = 336\ \mu\Omega$ cm). In **a–c**, the solid curve shows the $B = 0$ numerical renormalization group result for 1CK effect[59].

possible route to 2CK physics[3,4]. Theoretical arguments have, however, made it clear that this is not a viable route to nonmagnetic Kondo scattering[22,38]. Moreover, the creation of scattering centers in a real quantum material necessarily places the system in the weakly disordered regime where a conductance anomaly, the Altshuler–Aronov correction, occurs whose $T$ dependence can be mistaken for a 2CK signature, see, e.g., refs. [39–42]. Dilution studies on common Kondo lattice systems[43,44], on the other hand, typically create disorder distributions of Kondo temperatures that may result in a behavior of observables, which can easily be mistaken for that of a generic non-Fermi liquid[45].

We have shown that the low-$T$ resistivity anomaly in the transition-metal rutile IrO₂ is caused by $V_O$'s, demonstrating key signatures of an orbital 2CK effect and ruling out alternative explanations due to, e.g., the EEI effect. The most convincing argument in favor of 2CK physics would be the demonstration of direct tunability of 2CK physics to 1CK physics upon breaking the channel degeneracy. This is difficult, as the channel degeneracy is protected by time-reversal symmetry. A perhaps less direct, yet complementary, demonstration of this tunability is provided by our results for RuO₂ NWs which develop an orbital 1CK effect. In RuO₂, the antiferromagnetic order breaks the channel degeneracy. Our analysis also indicates that the underlying symmetries which support the existence of DNLs in the Brillouin zones of both transition-metal rutiles also aid the formation of orbital 2CK and 1CK physics.

Materials condensing in the rutile structure type and its derivatives form an abundant and important class that has helped shaping our understanding of correlated matter. The metal-insulator transition in VO₂, e.g., has been known for 60 years[46],

yet its dynamics is still not fully understood[47]. The demonstration that the non-symmorphic rutile space group supports a $V_O$-driven orbital Kondo effect in $M$O₂ holds promise for the realization of novel states of matter. The potential richness of orbital Kondo physics, e.g., on superconducting pairing, was recently pointed out in ref. [37] but may be even richer when considering the possibility of its interplay with topological band structures. Specifically, we envision the creation of a 2CK non-symmorphic superlattice of $V_O$'s in IrO₂ where the 2CK Majorana modes entangle with the band structure-enforced Dirac excitations forming a strongly correlated topological non-Fermi liquid state. Understanding its properties will foster deeper insights into the interplay of topology with strong correlations beyond the usual mean field treatment. The theoretical approach to this non-symmorphic superlattice is reminiscent of the topologically garnished strong-coupling fixed-point pioneered in the context of Weyl–Kondo semimetals[11,48], suitably generalized to capture the intermediate coupling physics of the 2CK effect and its low-$T$ excitations. The fabrication of superlattices of Kondo scattering centers has already been demonstrated[49] while defect engineering of vacancy networks, including $V_O$ networks is currently explored in a range of materials[50,51]. The specifics of this unique state and its manufacturing are currently being explored.

## Methods

**NW growth**. IrO₂ NWs were grown by the metal-organic chemical vapor deposition method, using (MeCp)Ir(COD) supplied by Strem Chemicals as the source reagent. Both the precursor reservoir and the transport line were controlled in the temperature range of 100–130 °C to avoid precursor condensation during the vapor-phase transport. High purity O₂, with a flow rate of 100 sccm, was used as the carrier gas and reactive gas. During the deposition, the substrate temperature

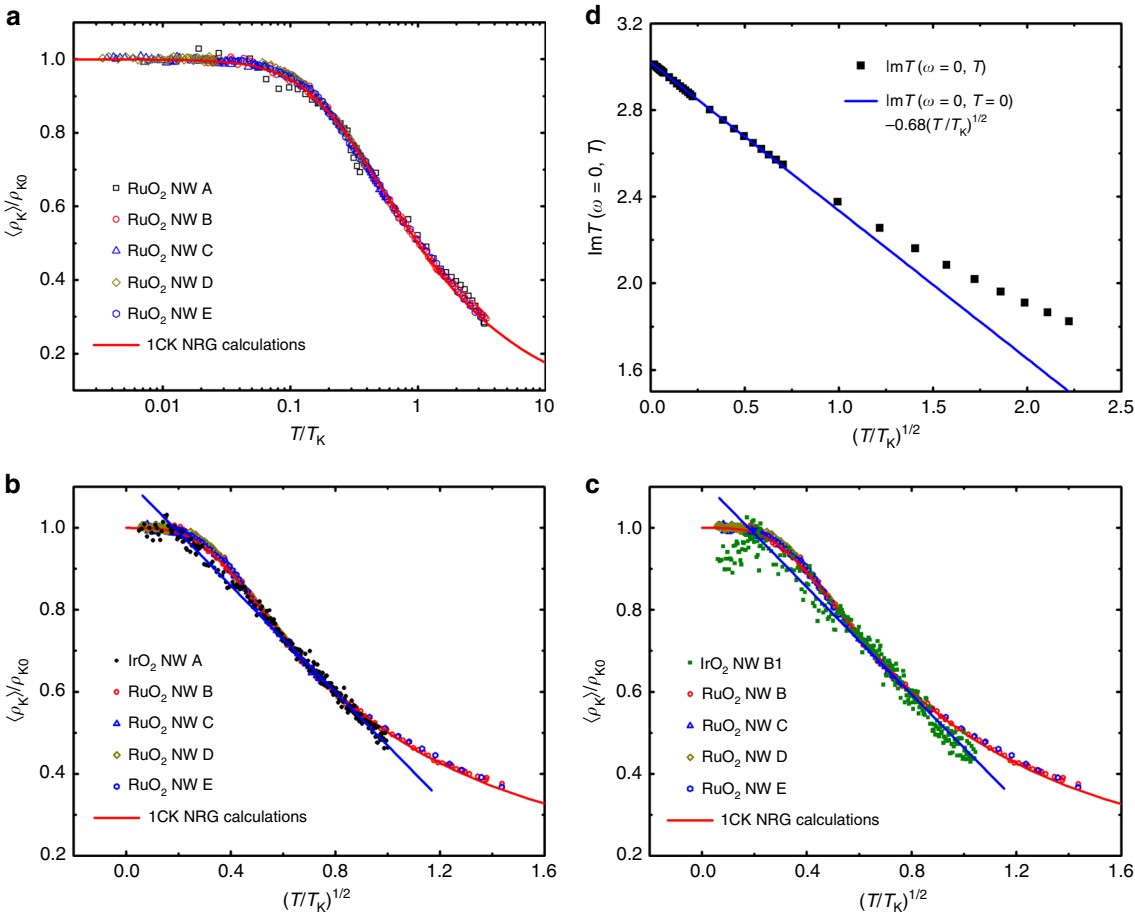

**Fig. 4 Comparison of 2CK and 1CK resistivities. a** Normalized Kondo resistivity $\langle\rho_K\rangle/\rho_{K0}$ versus $T/T_K$ for RuO$_2$ NWs A–E manifests the 1CK scaling form (solid curve) for over three decades of reduced temperature. **b** $\langle\rho_K\rangle/\rho_{K0}$ versus $\sqrt{T/T_K}$ for IrO$_2$ NW A and RuO$_2$ NWs B–E. The data of IrO$_2$ NW A obeys a $\sqrt{T}$ law between 0.39 and 21 K. For clarity, the experimental data points for RuO$_2$ NWs are plotted with small open symbols. **c** $\langle\rho_K\rangle/\rho_{K0}$ of IrO$_2$ NW B1 obeys a $\sqrt{T/T_K}$ law between 0.66 and 22 K, distinctively deviating from the 1CK function. **d** Results for the resistivity of a diluted system of 2CK impurities in a metallic host evaluated using a dynamical large-$N$ limit (black symbols), which follows a $\sqrt{T/T_K}$ law at low $T$ (see text and Supplementary Note 5). The ordinate is plotted in unit of half-bandwidth $W = 4$ eV (ref. [60]).

was kept at ≈350 °C and the chamber pressure was held at ≈17 torr to grow NWs[52,53]. Selected-area electron diffraction patterns[52] and X-ray diffraction (XRD) patterns[54] revealed a single-crystalline rutile structure.

RuO$_2$ NWs were grown by the thermal evaporation method based on the vapor-liquid-solid mechanism, with Au nanoparticles as catalyst. A quartz tube was inserted in a furnace. A source material of stoichiometric RuO$_2$ powder (Aldrich, 99.9%) was placed in the center of the quartz tube and heated to 920–960 °C. During the NW growth, an O$_2$ gas was introduced into the quartz tube and the chamber was maintained at a constant pressure of ≈2 torr. Silicon wafer substrates were loaded at the downstream end of the quartz tube, where the temperature was kept at 450–670 °C (ref. [55]). The morphology and lattice structure of the NWs were studied using XRD and high-resolution transmission electron microscopy (HR-TEM). The XRD patterns demonstrated a rutile structure[55], and the HR-TEM images revealed a polycrystalline lattice structure[56].

**Electrical measurements**. Submicron Cr/Au (10/100 nm) electrodes for 4-probe $\rho(T)$ measurements were fabricated by the standard electron-beam lithography technique. The electrode fabrication was done after the thermal treatment (annealing and/or oxygenation) of each NW was completed. To avoid electron overheating, the condition for equilibrium, $eV_s \ll k_BT$, was assured in all resistance measurements[57], where $e$ is the electronic charge, and $V_s$ is the applied voltage across the energy relaxation length. The electrical-transport measurements were performed on a BlueFors LD-400 dilution refrigerator equipped with room-temperature and low-temperature low-pass filters. The electron temperature was calibrated down to ≲50 mK. In several cases (RuO$_2$ NWs B–E), the measurements were performed on an Oxford Heliox $^3$He cryostat with a base temperature of ≃250 mK. The magnetic fields were supplied by superconducting magnets and applied perpendicular to the NW axis in all cases.

## Data availability
All data collected or analyzed during this study is available in the main text or the Supplementary Information material.

## Code availability
Details on the numerics is available upon request from the authors.

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

## Acknowledgements

The authors are grateful to F. R. Chen, J. J. Kai, and the late Y. S. Huang for growing $RuO_2$ and $IrO_2$ NWs, S. P. Chiu for experimental assistance, T. A. Costi for providing the 1CK scaling curve from NRG calculations, and Q. Si, A. M. Chang, C. H. Chung, and S. Wirth for helpful discussions. Figure 1a was produced with the help of VESTA[58]. This work was supported by Ministry of Science and Technology, Taiwan (grant Nos. MOST 106-2112-M-009-007-MY4, 108-3017-F-009-004, and 108-2811-M-009-500) and the Center for Emergent Functional Matter Science of National Chiao Tung University from The Featured Areas Research Center Program within the framework of the Higher Education Sprout Project by the Ministry of Education (MOE) in Taiwan. F.Z. and J.K. acknowledge financial support by the Deutsche Forschungsgemeinschaft (DFG) through SFB/TR 185 (277625399) and the Cluster of Excellence ML4Q (390534769). Work at Zhejiang University was in part supported by the National Key R&D Program of the MOST of China, grant No. 2016YFA0300202 and the National Science Foundation of China, grant No. 11774307.

## Author contributions

S.S.Y., S.K., and J.J.L. conceived the experiment. S.S.Y. and A.S.L. carried out electrical-transport measurements. T.K.S., C.C.L., and A.S.L. fabricated 4-probe NW devices with thermal treatments. F.Z. performed dynamical large-N calculations. J.K. provided theoretical support. S.S.Y., S.K., and J.J.L. analyzed and explained the data, and wrote the paper.

## Competing Interests

The authors declare no competing interests.
