## [Peer Review File · Nature Communications]

REVIEWER COMMENTS

Reviewer #1 (Remarks to the Author):

In the present study, the authors discuss the role of the Oxygen vacancies in topological metals IrO_2 and RuO_2 and claim that the low resistivity behaviour can be explain in terms of Kondo physics. They measure the resistivity as function of temperature and in the case of IrO_2 the resistivity scales as $\rho(T) \propto \sqrt{T}$ indicating a possible two channel Kondo (2CK) physics, while in the case of RuO_2 a single channel (1CK) behaviour is recovered. When scaled together, the resistivity from various samples for RuO_2 follows the 1CK resistivity curve as obtained with the NRG calculations. The experimental data if further compare with the theoretical results obtained using the Dynamical large-N approach.

It is well know that obtaining a 2CK state in mesoscopic devices requires allot of tuning, and under this circumstances, if the interpretation of the present data is correct, such a system will represent a robust realisation of the 2CK state. I found this the main take away message of the present work.

The results are therefore interesting and deserve publication in Nature Communication provided that the authors improve the manuscript along the following lines:

1. On the experimental side, when measuring the Kondo contribution to the resistivity in a regular metal (for example Cu or Au crystals doped with Fe), various concentration of impurity ions are considered and a systematic increase of the resistivity is observed with increasing doping. That is a guarantee that indeed the magnetic impurities do contribute to the resistivity. In the present study, there is no such an analysis with respect to the oxygen vacancies. That would add a major contribution to the present work and the message would be more clear.
2. It looks like the authors have a very good control on the crystallographic structure of their compounds. In that regard, the Kondo temperature, which is directly related (although exponentially dependent) to the exchange coupling of the impurity spin (in this case the electron at the site M_2 for example and the nearest-neighbors O_2 sites. To get a real 2CK state we need a perfect symmetrical coupling with the two bands. How is possible to have such a symmetrical situation but the values of the Kondo temperatures to be drastically different from one sample to the other?
3. In Fig. 3c) the authors display the magnetic field dependence. It is true that, the magnetic field should have no observable influence on the resistivity, which is in favour of the physical picture the authors provide. However, do the authors envision any mechanics (by applying strain for example) to deform the perfectly symmetric coupling and the 2CK behaviour to turn into a 1CK?

Reviewer #2 (Remarks to the Author):

The article "Oxygen vacancies in the topological metals IrO₂ and RuO₂: a simulator for strong electron correlations" report resistivity measurements in IrO₂ and RuO₂ nanowires. From their temperature and magnetic field dependence, the authors claim that they observe one and two channels orbital Kondo physics.

Unambiguous evidence of this high-correlated states in real materials have never been obtained.

IrO₂ is a good candidate to present a two-channel orbital Kondo state (2CK) due to the specificity of its band structure near oxygen vacancies and a high-symmetry protection.

RuO₂ is a good counterpart to IrO₂ presenting a similar crystalline structure but without the time-reversal symmetry and thus predicted to host a one channel Kondo state (1CK).

Yet I'm not fully convinced that these states are really observed in this article and won't recommend it for publication in Nature Communications in its present form.

I detailed below my main concerns:

1. First, I was confused about the Kondo effect description reading up to page 4. I was looking for the impurity spin and the screening channels having in mind the standard orbital Kondo picture of Nozières and Blandin (reference 2). The reference 4 describing the particular dynamical Kondo effect of the article is too confidential and not cited correctly here (only to claim that Kondo effect was not seen in real materials in the abstract and introduction). I think it should be clearly stated in the introduction that the invoked Kondo mechanism here is not the "standard" one. The mechanism should be precisely described before going into the details of the oxygen vacancies and the orbital channels. That would be more illuminating than talking about Majorana's modes.

2. The results for 1CK resistivity are supported by NRG calculation. Why are there no NRG calculations for 2CK? It is known that the large N- limit calculations in Kondo effects does not directly expand to the special case N=2 for all observables (see e.g. reference 7 where a linear dependence of conductance is found for a 2CK state). Could the authors comment on the degree of certainty of the expected temperature power law for the resistivity in the 2CK case?

3. The square root temperature dependence of the resistivity of IrO₂ nanowire below 20K is here demonstrated. Other mechanisms like electron-electron interactions leading to the same power law are shortly mentioned in the main paper and the useful discussions to rule out their role is only in the supplementary.

3.1. I think this discussion should be in the main paper.

3.2. The main argument to rule out electron-electron interactions in IrO₂ is the amplitude of the effect calculated from reference 23. First, it was already obtained experimentally, in standard diffusive metal wire, an amplitude larger by one order of magnitude than the calculated one (see e.g Phys. Rev. Lett. 90,076806 (2003)).

Second, it depends largely on the experimental parameters like the diffusion constant, the density of state at the Fermi level, the mean free path. I don't know how the diffusion constant of 6.3 cm²/s for nanowire B1 was inferred in the supplementary. What is the diffusion constant for all other wires? What is the density of states at the Fermi level? All these figures and the corresponding calculated amplitude should appear.

3.3. The nanowire dimensionality regarding electron-electron interactions depend not only on the thermal energy but also on the Thouless energy. I had to go down to the full supplementary to have a hint about the nanowires diameters. It appears that RuO₂ nanowires are about 5 times smaller than the IrO₂ wire. Can this have an influence on the different power laws? This should be discussed (even if no linear temperature dependence is predicted due to electron electron interactions in this case).

4. Other B-independent mechanisms leading to temperature power laws of resistivity in one dimension like dynamical Coulomb blockade, Luttinger liquid with impurities are not ruled out and mentioned.

Oxygen vacancies in the topological metals IrO₂ and RuO₂: a simulator for strong electron correlations by Sheng-Shiuan Yeh et al.

We are grateful to both referees for their careful reading of our manuscript and their constructive and professional feedback. Below, we address each of the points raised by the referees and outline the resulting changes to the manuscript. The modifications appear in blue in the updated manuscript and supplement.

ANSWER TO REFEREE 1:

We thank the referee for the careful reading of the manuscript, for helpful feedback and encouraging comments as well as for supporting publication in Nature Communications.

The results are therefore interesting and deserve publication in Nature Communication provided that the authors improve the manuscript along the following lines:

1. On the experimental side, when measuring the Kondo contribution to the resistivity in a regular metal (for example Cu or Au crystals doped with Fe), various concentration of impurity ions are considered and a systematic increase of the resistivity is observed with increasing doping. That is a guarantee that indeed the magnetic impurities do contribute to the resistivity. In the present study, there is no such an analysis with respect to the oxygen vacancies. That would add a major contribution to the present work and the message would be more clear.

Ans: Linking the observed low-temperature (low- T) resistivity anomaly to oxygen defects is indeed very important. In the original submission, we discussed the link between the low- T resistivity increase and the presence of oxygen vacancies. We demonstrated that in fully oxygenated nanowires (IrO₂ NW 3 in the inset of Fig. 2, and RuO₂ NW 4 in the inset of Fig. 3c and NWs F and G in Fig. S2b), no low- T resistance increase was found. Then, we showed that, in those nanowires annealed in vacuum to generate oxygen vacancies, a low- T resistivity increase was observed (IrO₂ NWs A and B1 in Fig. 2). Furthermore, we had aged NW B1 in air for 5 months to absorb oxygen. The resistivity increase in this case (referred to as NW B2) still persisted, but the magnitude was smaller than that in NW B1, due to the reduced oxygen vacancy density. Similarly, RuO₂ NW A was annealed in argon, which resulted in a resistivity increase (Fig. 3c). These thermal annealing experiments thus establish that the observed low- T resistivity anomaly is caused by the presence of oxygen vacancies.

We already established in Ref. [24] (all citation numbers refers to the updated manuscript) that oxygen vacancies in MO₂ can be reversibly generated (removed) by thermal annealing in vacuum or an argon gas (an oxygen gas). Systematic $1/f$ noise and X-ray photoelectron spectroscopy (XPS) studies were carried out to confirm this assertion, as well as to extract the number density of oxygen vacancies (n_{V_O}) from the measured $1/f$ noise magnitudes.

We have extracted the n_{V_O} values in the MO₂ nanowires, based on the available theoretical expressions of Eqs. (S8) and (S9). The resulting values are listed in Table 1 (originally Table S1). These values are in line with those values extracted from $1/f$ noise studies [24].

In the original manuscript this information may not have been as prominently featured as it should have. In the revised version, we explicitly state in the main text the annealing and/or oxygenation conditions. This revision should make it much clearer that a low- T resistivity increase is only found in those nanowires containing oxygen vacancies. As for the relation between the resistivity and the concentration of orbital Kondo scatterers n_{V_O} , we note that, with the exception of NW B, our data indeed indicate an approximately linear relation between n_{V_O} and the residual resistivity ρ_{B0} for RuO₂ nanowires. (We focus on RuO₂ nanowires because of the larger number of samples and the larger variation of ρ_{B0} values). It is not unexpected that the approximately linear relation between ρ_{B0} and n_{V_O} holds for larger impurity concentrations, corresponding to larger values of ρ_{B0} as all defects, screened dynamic and static defects contribute to ρ_{B0} . (NW B was taken from the same batch as other RuO₂ nanowires. Probably due to its position located on the silicon wafer substrate during growth processes, or due to the submicron electrode fabrication processes, it contained a somewhat larger amount of oxygen vacancies than expected.) We have added the discussion and a new figure (Fig. S3) for n_{V_O} versus ρ_{B0} to Supplement S2.

2. It looks like the authors have a very good control on the crystallographic structure of their compounds. In that regard, the Kondo temperature, which is directly related (although exponentially dependent) to the exchange coupling of the impurity spin (in this case the electron at the site M2 for example and the nearest-neighbors O2 sites). To get a real 2CK state we need a perfect symmetrical coupling with the two bands. How is possible to have such a symmetrical situation but the values of the Kondo temperatures to be drastically different from one sample to the other?

Ans: The observed variation of the Kondo temperature in the IrO₂ nanowires is negligible but the one in RuO₂ nanowires is indeed substantial, see Table 1. The variations of the Kondo temperature in RuO₂ nanowires implies a variation in $J_K\rho(E_F)$, where J_K is the ‘orbital’ exchange coupling and $\rho(E_F)$ is the local density of states of conduction electrons that couple to the orbital degree of freedom. As the referee pointed out, the dependence of the Kondo temperature on this quantity is exponential. A small variation of J_K could explain the variation in the Kondo temperature as long as this variation would leave the orbital degeneracy intact while the channel symmetry in IrO₂ is ensured by time-reversal symmetry. More likely, however, is a variation in $\rho(E_F)$. It was shown by S.-S. Yeh et al. that RuO₂ nanowires contain large amounts of nanograins with sizes ranging from a characteristic length of a few to several tens of nanometer [57]. The number and size distribution of these nanograins are dependent on the diameter of the nanowire and on the overall growth processes and lead to a slightly different averaged $\rho(E_F)$ for each RuO₂ nanowire. As already discussed in the Supplement S7 of the original manuscript, the theory of Chakravarty and Nayak then answers why a single Kondo temperature is observed in each nanowire.

3. In Fig. 3c) the authors display the magnetic field dependence. It is true that, the magnetic field should have no observable influence on the resistivity, which is in favour of the physical picture the authors provide. However, do the authors envision any mechanics (by applying strain for example) to deform the perfectly symmetric coupling and the 2CK behaviour to turn into a 1CK?

Ans: We fully agree with the referee. Strain and shear are expected to deform the structure and thereby lift the orbital degeneracy. This would be the analog of a magnetic field in the standard spin Kondo model. We have plans to explore the effect of strain on the orbital Kondo effect and have already started to design an experiment to perform this kind of study. This probably can be achieved by artificially unflattening the substrate. However, we should point out that our nanowires are about 1 micrometer long. Therefore, to apply an external strain precisely at the nanowire position and which is large enough to deform the symmetric coupling would be a nontrivial task. Moreover, strain will per se not break the time-reversal symmetry which protects the channel degeneracy including the orbital exchange couplings. Instead, for sufficiently large strain, the resulting gap between the two orbital states will simply suppress Kondo scattering. In any case, we will proceed measurements in this direction.

ANSWER TO REFEREE 2:

We thank the referee for the careful reading of the manuscript as well as for her/his constructive comments and helpful feedback. Below, we address the questions and suggestions of the referee and indicate the changes made to the manuscript to accommodate them.

I detailed below my main concerns:

1. First, I was confused about the Kondo effect description reading up to page 4. I was looking for the impurity spin and the screening channels having in mind the standard orbital Kondo picture of Nozières and Blandin (reference 2). The reference 4 describing the particular dynamical Kondo effect of the article is too confidential and not cited correctly here (only to claim that Kondo effect was not seen in real materials in the abstract and introduction). I think it should be clearly stated in the introduction that the invoked Kondo mechanism here is not the "standard" one. The mechanism should be precisely described before going into the details of the oxygen vacancies and the orbital channels. That would be more illuminating than talking about Majorana’s modes.

Ans: In light of the referee’s comments, we realize that the readability of our manuscript had considerable room for improvement. Thus, we now state explicitly at the beginning of the manuscript that we establish the realization of two- and one-channel orbital Kondo physics in IrO₂ and RuO₂ nanowires and briefly introduce the basic idea of this realization. We also briefly touch on the history of non-magnetic Kondo physics which, at least as a theoretical possibility, has been around for many decades. Moreover, we also added the term ‘non-magnetic’ to the abstract. Regarding Ref. [38] (originally Ref. [4]) we do not consider it to be (too) confidential. It is true that Ref. [38] makes the point that there are no conclusive observations of the orbital Kondo effect in real quantum materials but this paper also points to the rich potential of the orbital Kondo effect to induce new states of matter including, *e.g.*, odd-frequency pairing. Since we envision that transition metal oxides possessing the rutile structure will allow us to explore symmorphic and non-symmorphic orbital Kondo lattices, we believe it is appropriate to refer to Ref. [38]. In order to accommodate the referee’s suggestions while retaining the original discussion, we have move part of the introduction, including reference to [38], to the DISCUSSION section. We have also removed the reference to

Majorana fermions from the abstract.

2. The results for 1CK resistivity are supported by NRG calculation. Why are there no NRG calculations for 2CK? It is known that the large N-limit calculations in Kondo effects does not directly expand to the special case N=2 for all observables (see e.g. reference 7 where a linear dependence of conductance is found for a 2CK state). Could the authors comment on the degree of certainty of the expected temperature power law for the resistivity in the 2CK case?

Ans: We thank the referee for asking us to clarify this point. In general, the singular temperature dependence in the vicinity of the multi-channel Kondo fixed point is determined by the leading irrelevant operator. For the magnetic two-channel Kondo problem possessing $U(1) \times SU(2) \times SU(2)$ symmetry, its effect on the local one-particle Green's function has been calculated by Affleck and Ludwig [33] and leads to the well-known (low- T) \sqrt{T} behavior for the conduction electron scattering T-matrix. The $U(1) \times SU(2) \times SU(2)$ symmetry reflects charge conservation and spin and orbital symmetry. On general grounds, enhancing the symmetry is not expected to lead to additional irrelevant operators, while a symmetry reduction has the potential to change the singular temperature dependence of observables. Nonetheless, even a two-level system immersed in a metal would display a \sqrt{T} contribution to the resistivity if such a system could be placed near the two-channel Kondo fixed point. In this case, the Kondo-active degree of freedom would only possess an easy-plane symmetry [3,22]. In our case, the role of spin and orbital degrees of freedom are interchanged and no more singular leading irrelevant operator is created [4]. Thus, a \sqrt{T} resistivity behavior is indeed expected and observed. The dynamical large-N method is closely related to the non-crossing approximation (NCA) and captures the leading singular behavior even at N=2, *i.e.*, for the two-channel Kondo model, see Refs. [34-36]; we have added a corresponding statement to the end of the RESULTS section. The method, however, does fail for the one-channel case where a Fermi liquid ground state ensues. In general, NRG is a very powerful method for addressing general quantum impurity problems. The dynamical large-N limit also works well for the two-channel Kondo problem and can even work in cases where the NRG will face difficulties, *e.g.*, due to the logarithmic discretization or beyond the linear response regime. The referee correctly points out that the realization of two-channel Kondo physics demonstrated in Refs. [7,8] displays a linear-in-temperature conductance. This realization is based on a proposal by K. A. Matveev (K. Matveev, JETP **72**, 892 (1991)). Unlike the magnetic or orbital two-channel Kondo effect, in this realization, it is the charging state of the quantum dot that plays the role of the Kondo-active degree of freedom. In that case, the capacitance is the analog of the spin (or in our case orbital) susceptibility. Moreover, this particular realization is placed right at the so-called Emery-Kivelson point, where the leading irrelevant operator decouples from the problem (V. J. Emery and S. Kivelson, Phys. Rev. B **46**, 10812 (1992); and A. M. Sengupta and A. Georges, Phys. Rev. B **49**, 10020 (1994)). As a result, one finds a linear-in-temperature conductance [7,8] (see also A. Furusaki and K. A. Matveev, Phys. Rev. B **52**, 16676 (1995)).

3. The square root temperature dependence of the resistivity of IrO₂ nanowire below 20K is here demonstrated. Other mechanisms like electron-electron interactions leading to the same power law are shortly mentioned in the main paper and the useful discussions to rule out their role is only in the supplementary.

3.1. I think this discussion should be in the main paper.

Ans: As suggested by the Referee, we have moved the discussion regarding the electron-electron interaction (EEI) from the original Supplementary S3 to the main text. In order not to divert the readers' attention from the main result of this work, *i.e.*, the observation of 2CK physics in a real quantum material, we have, however, retained the numerical estimates with additional reasons for ruling out the EEI effect in the Supplement S3.

3.2. The main argument to rule out electron-electron interactions in IrO₂ is the amplitude of the effect calculated from reference 23. First, it was already obtained experimentally, in standard diffusive metal wire, an amplitude larger by one order of magnitude than the calculated one (see e.g. Phys. Rev. Lett. **90**,076806 (2003)). Second, it depends largely on the experimental parameters like the diffusion constant, the density of state at the Fermi level, the mean free path. I don't know how the diffusion constant of 6.3 cm²/s for nanowire B1 was inferred in the supplementary. What is the diffusion constant for all other wires? What is the density of states at the Fermi level? All these figures and the corresponding calculated amplitude should appear.

Ans: First, in Phys. Rev. Lett. **90**, 076806 (2003), the electron inelastic scattering rate was found in one of the diffusive metal wires which contained dilute magnetic impurities. As a result, the large quasiparticle energy relaxation rate was caused by magnetic-impurity-mediated interactions, but not due to enhanced EEI. Therefore, this is a situation distinct from the present observation of the nonmagnetic orbital Kondo effect. Moreover, the size of the amplitude of the low- T resistivity anomaly is not our only argument to rule out the EEI effect. The deviation from the \sqrt{T} behavior observed in our IrO₂ nanowires at the lowest temperatures is also incompatible with the EEI effect

as the underlying cause. The EEI effect is magnetic field dependent unless the screening parameter (\tilde{F} in Eq. (S1)) identically vanishes [26] which would have to occur in all our NWs. Finally, the EEI effect cannot even in principle account for the low- T resistivity anomaly observed in RuO₂ nanowires.

Second, in this revised manuscript, we have moved the original Table S1 to the main text (now renamed as Table 1). In the Table, we have added a new column to list the diffusion constant for every nanowire. In the table caption, we have added the expression for calculating the diffusion constant: $D = 1/[\rho e^2 N(E_F)] = v_F \ell / 3$.

3.3. The nanowire dimensionality regarding electron-electron interactions depend not only on the thermal energy but also on the Thouless energy. I had to go down to the full supplementary to have a hint about the nanowire diameters. It appears that RuO₂ nanowires are about 5 times smaller than the IrO₂ wire. Can this have an influence on the different power laws? This should be discussed (even if no linear temperature dependence is predicted due to electron-electron interactions in this case).

Ans: We have moved Table 1 to the main text to make the information more readily available. We also stress now in the main text that our nanowires, regardless of their diameters, are all three-dimensional (3D) with respect to Boltzmann transport. Moreover, the thermal diffusion length $L_T = \sqrt{D\hbar/k_B T}$ is short compared to the nanowire diameter (for example, $L_T = (63 \pm 6)/\sqrt{T}$ nm in IrO₂ nanowires). Therefore, the EEI effect, if present, are 3D and thus should display a \sqrt{T} dependence. This \sqrt{T} contribution would, however, be more than one order of magnitude smaller than that observed in Fig. 2. In RuO₂ nanowires, $L_T = (26 - 57)/\sqrt{T}$ nm is also short. Moreover, we do not observe any \sqrt{T} dependence in Fig. 3. Thus, the differences between IrO₂ and RuO₂ cannot be due to the different dimensions of our nanowires. On the other hand, the 1CK scaling over three decades of reduced temperature (Fig. 4a) strongly rules out the 3D EEI effect. These points have been stressed where appropriate in the revised manuscript. In our MO₂ nanowires, the relation between the various energy scales are: inelastic level broadening $\sim k_B T \gg$ Thouless energy ($= D\hbar/L^2$, where $L \approx 1 \mu\text{m}$ is the nanowire length) \gg energy level spacing. That is, our nanowires fall in the 3D continuous spectrum limit.

4. Other B-independent mechanisms leading to temperature power laws of resistivity in one dimension like dynamical Coulomb blockade, Luttinger liquid with impurities are not ruled out and mentioned.

Ans: We agree with the referee that both phenomena mentioned, *i.e.*, dynamical Coulomb blockade physics and impurity physics in one-dimensional metals can lead to a power law behavior of the resistivity. We can, however, with confidence rule out both phenomena as possible sources for the observed power law behavior. Dynamical Coulomb blockade, *e.g.*, requires contacts with small tunneling conductances to observe the resulting power law behavior in temperature and bias voltage across the junction. The resulting power law exponent is given by the resistance in units of the quantum of resistance, $h/(2e^2)$ [S15]. Changes in the diameter of a quasi one-dimensional conductor affect the number of transmission channels coupled to the contact and thus the dynamical Coulomb blockade exponent in contrast to the universal exponent we observe. Moreover, the fundamentally different behavior of the behavior in RuO₂ nanowires from that of IrO₂ nanowires cannot be explained by dynamical Coulomb blockade. Finally, as our nanowire diameters are much larger than the electron mean free paths, the electrical transport is 3D and well described in terms of Boltzmann transport with disorder-induced resistivity corrections at low T . This 3D nature also rules out impurities in Luttinger liquids as a possible cause for the observed power law behavior. In addition, the T dependence of the resistivity in this case would have a complicated and impurity concentration dependent structure [S16] in contrast to our observations. The low- T transport anomalies which we report are intrinsic properties of IrO₂ and RuO₂. The use of nanowires ensures that the relative resistance change at (small) current densities that avoid electron heating is detectable. This was previously discussed in METHODS and the Supplement S3. In the updated version, this information can be found in the main text.

REVIEWERS' COMMENTS:

Reviewer #1 (Remarks to the Author):

The revised manuscript is an improved version of the previous one, where the authors have included new paragraphs where they discuss the issues raised by both Referees. In the new manuscript the authors answers to all my concerns that I had raised in the first report and implemented the changes accordingly.

The paper is now clearly written, the new introduction where the discussion is focused on the Kondo physics -along the line suggested by the second Referee- being more suitable for the present work. By the new changes the authors harmonised parts of the works and to me the paper reads nicely.

I have no further concerns and I believe that this new version is suitable for publication in Nature Communications in its present form.

Reviewer #2 (Remarks to the Author):

I thank the authors for their answers and the modification of their article. I now find it more readable and more convincing.

Yet I still think that the word "unambiguous" used in the abstract ("We report the unambiguous observation of orbital one- and two-channel Kondo physics") and added in the first sentence of the paragraph Experimental signatures of orbital 2CK effect is too strong.

Indeed as stated by the author at the bottom of page 6 they have to rule out electron electron interactions. This is done here:

- by calculating the prefactor with a formula which holds for weakly disordered metal so the product $kF l$ is crucial. They might be deviation at the frontier of this regime close to the experimental parameters.
- by claiming that this prefactor would depend on the magnetic field but this dependence is not expected to appear here with the factor $F \sim 0.1$.
- by comparing with RuO₂ wire where no square root dependence is seen. Could this dependence be screened by the 1CK effect?

In conclusion, I think that the authors might have observed two channel Kondo effect in real materials but that they have to be cautious and remove the unambiguous term.

SECOND REMARKS BY/ANSWER TO REFEREE 1:

The revised manuscript is an improved version of the previous one, where the authors have included new paragraphs where they discuss the issues raised by both Referees. In the new manuscript the authors answers to all my concerns that I had raised in the first report and implemented the changes accordingly.

The paper is now clearly written, the new introduction where the discussion is focused on the Kondo physics -along the line suggested by the second Referee- being more suitable for the present work. By the new changes the authors harmonised parts of the works and to me the paper reads nicely.

I have no further concerns and I believe that this new version is suitable for publication in Nature Communications in its present form.

Ans: We thank the referee for carefully reading our revised manuscript and for supporting its publication in Nature Communications.

SECOND REMARKS BY/ANSWER TO REFEREE 2:

I thank the authors for their answers and the modification of their article. I now find it more readable and more convincing.

Yet I still think that the word “unambiguous” used in the abstract (“We report the unambiguous observation of orbital one- and two-channel Kondo physics”) and added in the first sentence of the paragraph Experimental signatures of orbital 2CK effect is too strong.

Indeed as stated by the author at the bottom of page 6 they have to rule out electron electron interactions. This is done here:

- by calculating the prefactor with a formula which holds for weakly disordered metal so the product $k_F \ell$ is crucial. They might be deviation at the frontier of this regime close to the experimental parameters.

- by claiming that this prefactor would depend on the magnetic field but this dependence is not expected to appear here with the factor $F \sim 0.1$.

- by comparing with RuO₂ wire where no square root dependence is seen. Could this dependence be screened by the 1CK effect?

In conclusion, I think that the authors might have observed two channel Kondo effect in real materials but that they have to be cautious and remove the unambiguous term.

Ans: We thank the referee for carefully reading our manuscript and for supporting its publication in Nature Communications. Following the referee’s request, we have deleted the word “unambiguous” in the abstract. In addition, in the first sentence of the paragraph Experimental signatures of orbital 2CK effect, we have replaced the wording “the unambiguous realization of ...” by “the most convincing realization of ...”.